# Conversion to secondary progressive multiple sclerosis: Multistakeholder experiences and needs in Italy

**Ambra Mara Giovannetti**[1,2], **Erika Pietrolongo**[3], **Claudia Borreani**[4], **Andrea Giordano**[1,5], **Insa Schiffmann**[6], **Anna Barabasch**[6], **Christoph Heesen**[6], **Alessandra Solari**[1]\*, **for the ManTra Project**[¶]

1 Unit of Neuroepidemiology, Fondazione IRCCS Istituto Neurologico Carlo Besta, Milan, Italy, 2 Unit of Neuroimmunology and Neuromuscular Diseases, Fondazione IRCCS Istituto Neurologico Carlo Besta, Milan, Italy, 3 Department of Neuroscience, Imaging and Clinical Sciences, G d'Annunzio University of Chieti-Pescara, Chieti, Italy, 4 Unit of Clinical Psychology, Foundation IRCCS Istituto Nazionale per la Cura dei Tumori, Milan, Italy, 5 Department of Psychology, University of Turin, Turin, Italy, 6 Institute of Neuroimmunology and Multiple Sclerosis, University Medical Center Hamburg-Eppendorf, Hamburg, Germany

¶ Membership of the ManTra Projec is provided in the Acknowledgments.
\* alessandra.solari@istituto-besta.it

**Data Availability Statement:** All relevant data are within the manuscript and its Supporting Information files.

## Abstract

### Background

Conversion to secondary progressive multiple sclerosis (SPMS) is associated with a relatively poor prognosis, and SPMS is responsible for the majority of the social and economic costs associated with MS. Managing the Transition to SPMS (ManTra) is a mixed methods project conducted in Italy and Germany aimed to set up a user-led resource to empower and improve the quality of life of newly diagnosed SPMS patients.

### Aims

To assess the experiences and the needs of Italian people who recently converted to SPMS, patient significant others (SOs), neurologists and other health professionals (HPs).

### Methods

We conducted 15 personal semistructured interviews (PSIs) with SPMS patients who transitioned up to five years, and three focus group meetings (FGMs), one of SPMS SOs, one of neurologists, and one of other HPs. Participants were purposely selected from the three geographic areas of Italy, and varied in terms of gender, education and (for patients) disease severity. PSIs and FGMs were audiorecorded, transcribed and analyzed by two researchers using the framework analysis.

### Results

One hundred sub-categories were identified, grouped into 13 categories and four themes: 'awareness of the transition', 'communication of the transition', 'dealing with symptoms

**Funding:** This study is supported by the Fondazione Italiana Sclerosi Multipla (FISM, https://www.aism.it, grant N° 2015/R/22 to AS). The funding source had no role in study design, data collection, data analysis, data interpretation or report writing.

**Competing interests:** AS is affiilliated with Biogen Idec, Merck Serono, Novartis, Almirall, and Excemed. CH is affiliated with Biogen, Genzyme, Sanofi Aventis, Merck, Novartis, and Roche. There are no patents, products in development or marketed products to declare. This does not alter our adherence to PLOS ONE policies on sharing data and materials.

worsening', and 'needs'. The major unmet needs were collected in four dimensions 'organization and management; 'empowerment training'; 'information'; and 'policies'.

## Conclusions

Two are the main findings: first, the widespread lack of awareness around the transition; second, the need to improve the quality of the care pathway in the Italian context. It was particularly stressed the need for a holistic and multidisciplinary approach (with patients and SOs as members of the team), the development of an ad hoc plan of follow up visits with easy access to MS specialists' consultation/treatment; specialized training for each stakeholders; more information on SPMS, daily management and changes at policy level.

## Introduction

Multiple sclerosis (MS) is the most common cause of non-traumatic disability in young adults, with over 2.5 million sufferers worldwide, mainly women [1]. Demyelination and neurodegeneration are the hallmarks of the disease. Inflammatory demyelination causes episodes of neurological deficits that last for days or weeks, with complete or incomplete recovery (relapsing-remitting MS, RRMS). Axonal damage and neuronal loss underly the progressive accumulation of permanent clinical disability (secondary progressive MS, SPMS); in a minority of patients, a progressive course without relapses is present from clinical onset (primary progressive MS) [2].

SPMS is responsible for the majority of the social and economic costs associated with MS, and few partially effective disease-modifying treatments are currently available for this disease form [3]. Around 2–3% RRMS patients per year convert to SPMS [2], and 10–15 years after clinical onset half of the patients have developed SPMS in natural history studies of untreated cohorts [2, 4]. SPMS is diagnosed retrospectively after a period of diagnostic uncertainty that may last for several years, and neither imaging criteria nor biomarkers are available to distinguish RRMS from SPMS [5].

From a scoping review of the literature, we found three studies (four publications) on the experiences of patients around the RRMS–SPMS transition [6–9]. One additional study was published recently [10]. All the studies employed qualitative research, were published in the last decade and were conducted in the UK. Davies et al. identified four main themes envisaged by patients and carers: realization of the transition, reaction to this realization, realities of living with SPMS (dealing with the healthcare system in this period), and future challenges [6]. The same group also explored the experiences of HPs and found two main themes: recognizing and communicating about SPMS, and providing support [7]. Hourihan identified five themes from patient experiences: naming of the process of change, psychological consequences, consequences to occupations, impact on relationships, and coping with a life of change [8]. O'Loughlin et al. focused on patient and HP experiences, and suggested a process of moving from uncertainty toward confirmation of patient's diagnostic label, the experience of which was moderated by HP attitudes and approaches [9]. Bogosian et al. used a longitudinal design (two telephone interviews with patients one year apart) and found increased emotional and physical challenges after SPMS transition and between the first and second interview. Participants described feelings of helplessness, a lack of directions for symptoms management and lack of support from the healthcare system [10].

Managing the Transition to SPMS (ManTra) is a mixed methods project conducted in Italy and Germany that adheres to the Medical Research Council framework for developing and evaluating complex interventions [11]. The project goals were twofold: to assess the experiences and the needs of people who recently converted to SPMS, patient significant others (SOs), MS neurologists and other health professionals (HPs) and to set up a user-led resource to empower and improve the quality of life and autonomy of newly diagnosed patients with SPMS [12]. In line with the above studies, we were interested in assessing the experiences and needs of patients who recently converted to SPMS. Differently from them, we wanted to explore this topic in two different European countries, Italy and Germany, and we used both qualitative and quantitative methods. The present paper reports the results of the Italian qualitative findings.

## Methods

The experiences and needs of patients around the transition to SPMS was explored by personal semistructured interviews (PSIs) with recently diagnosed patients with SPMS, and by focus group meetings (FGMs) involving patient SOs, neurologists and other HPs involved in the MS care, as specified in the ManTra protocol [12].

The consolidated criteria for reporting qualitative research (COREQ) guided findings presentation [13, 14]. The adherence to the COREQ checklist is documented in S1 Appendix.

AG, AMG, EP and CB devised the PSI and FGM guides (S2 Appendix) with input from the Psychologist Working Group, five MS psychologists from various institutions, who participated in a four-day training course in qualitative research methods where the ManTra project was the "case study". No changes were needed in PSI and FGMs guides, after piloting them.

Participants' enrollment followed a purposive sampling technique. To cover a range of experiences, they were selected from the three geographic areas of Italy (North, Central, South), and varied in terms of gender, education and (for patients) disease severity (Expanded Disability Status Scale [EDSS] score) [15].

Patients (and SOs) were contacted by their neurologist and who showed interest in participating in the study were then contacted by the interviewer. HPs were invited by the study coordinator.

Before PSIs/FGMs began, participants were informed of study aims and requirement and signed the written consent, in accordance with the Helsinki Declaration and EU Good Clinical Practice guidelines. PSIs and FGMs were audiorecorded and verbatim transcribed. A complete audit trail of the study is reported in S3 Appendix.

### Interviews

A minimum of 15 PSIs was planned (five from each center/area of Italy), with sampling continuing until no new themes emerged from the narratives (data saturation) [16].

The interviews were conducted by a psychologist experienced in qualitative research at the MS Center of Milan (AMG), Chieti and Bari (EP).

The interviewer presented the purpose of the PSI and then posed the questions. Patients were encouraged to provide the interviewer with their opinion and perspective on the transition to SPMS and their needs.

Adults ($\geq$18 years of age) with SPMS were included if they were: diagnosed with SPMS [2] 3 months to 5 years before inclusion, fluent in Italian, and gave written consent. Patients with severe cognitive compromise (referring neurologist judgement) or unable to communicate effectively were excluded.

## Focus group meetings

Three FGMs were planned: one each for SOs, neurologists and other HPs working with people with SPMS.

Each FGM was planned to have 6 to 10 participants plus two moderators (AMG and EP) and run at the Fondazione IRCCS Istituto Neurologico C. Besta, Milan. The principal moderator first explained the aim of the meeting, and asked participants to introduce themselves. She then introduced each topic in turn, facilitating the discussion. After all pre-specified topics had been discussed, the moderator summarized the key points, and asked for further comments. The co-moderator took notes and oversaw the audio recording. By two weeks from the FGMs, participants received a report of the meeting for review (respondent validation).

**Eligibililty criteria.** SOs were relatives, partners or close friends of the patient who received a diagnosis of SPMS in the 3 months to 5 years prior to inclusion. SOs were eligible if they were ≥18 years of age, provided emotional or tangible support to the patient during the SPMS disclosure period [17], were fluent in Italian, and gave written consent. SOs were excluded if they had cognitive compromise or any impairment preventing effective communication.

Neurologists and other HPs were included if they had expertise in caring for SPMS patients, were fluent in Italian and provided written informed consent.

## Ethics statement

The protocol and consent procedures were approved by the Ethics Committee of Fondazione IRCCS Istituto Neurologico Carlo Besta, Milan (refs. no. 29, 04 May 2016; 43, 06 September 2017; 54, 12 September 2018); University G. d'Annunzio, Chieti (refs. no. 19, 03 November 2016); University of Bari, Bari (refs. no. 703, 19 October 2016).

## Analysis

The methods of framework analysis was applied to the PSI and FGM transcripts. Framework analysis uses an inductive approach to identify, extract and analyse core themes [16, 18, 19]. The transcripts were analysed in six successive steps (see below), each of which embodies an increasing level of generalization [20]. To enhance the validity of this process, two researchers analyzed the transcripts independently (FGMs, steps 1–4; PSIs, steps1–5) and jointly (step 6).

Steps in the analysis:

1. The researcher identifies all propositions considered significant, without considering their relation to other parts of the transcript and appends comments to these significant propositions.

2. Comments are expanded and contextualized along with the entire PSI/FGM.

3. Relations between comments are established by reordering and regrouping them by subject.

4. Themes are extrapolated and hierarchically ordered into categories, moving from general concepts to more specific ones.

5. Each PSI/FGM transcript analysis is compared with the others to identify differences and commonalities in the themes.

6. The analyses produced by the two researchers are compared, and a consensus is arrived at.

Once this process was completed, AMG and EP presented the results of the analysis that were discussed with the Qualitative Analysis Panel and the Psychologist Working Group.

## Results

### Participants and setting

Between March and August 2017, 15 interviews (mean duration 51.1 minutes; range 30–67) were conducted at the three participating MS centers. All the three FGMs took place between May and September 2017 at the Fondazione IRCCS Istituto Neurologico C. Besta (SOs, 107 minutes; HPs, 108 minutes; N, 109 minutes). One SO participated via audioconference. Participants' characteristics are reported in Table 1.

### Qualitative findings

From participants narratives, 100 sub-categories emerged which were grouped into 13 categories and 4 themes: 'awareness of the transition', 'communication of the transition', 'dealing with symptoms worsening', and 'needs' (see Tables 2–5). The findings are presented by theme, with only the most relevant quotes to illustrate their derivation. The provenance of quotes is indicated as patient (P), SO, neurologist (N) or HP, with other information included as appropriate (sex, age, EDSS score [15], relation to patient, profession, and center). The complete list of quotes for each sub-category is reported in S3 Appendix.

**Awareness of the transition.** Data showed a great variability in patients' awareness of their transition to SPMS (Table 2). Some reached this insight before they discussed it with the neurologist, other only after neurologist's disclosure, but a third group was totally unaware of it and someone realized it by participating in this project.

> *"I did not know even if they wrote it. Then I decided to let doctor support me, I trust them and I have always done what they told me to do. I did not ask for anything because when I knew I had MS, I panicked . . . I don't know where this situation will bring me, so I accepted to participate in a trial. I did not know what was that 'SP'. I only read the [magnetic resonance imaging, MRI] report that said the disease was still stable; just a new micro signal so, ok, let's go on!" [P02: woman, 60 years, EDSS 4, North]*

When this was the case, the interviewer had trouble in openly discussing with the patient about his/her experience and the interview was characterized by vagueness, predominance of interviewer verbalization, unclear language and verbosity, as reported in this interviewer's note:

> *"You can feel this person was unaware of the transition, because she could not stop "beating around the bush". She speaks a lot, in a verbose style, without the ability to reach the target. It seems like she wanted to fill in the conversation, but with few awareness of the topic."*

With the exception of those who discovered their transition to SPMS by participating in this project, awareness of the transition was mainly due to symptoms worsening despite MRI stability, lack of remissions, increasing difficulties in daily life, and discontinuation of disease modifying treatment.

> *"I knew my MS was a RR type, and my MRI shows no activity, but I can feel a slow worsening . . . It is only 8 months that I am using the wheelchair . . . I often asked my neurologist: How is it possible? My MRI is ok and I'm worsening." [P15: man, 39 years, EDSS 7, South]*

**Table 1. Characteristics of participants to personal semi-structured interviews (patients) and focus group meetings (patient significant others, SOs; neurologists; other health professionals, HPs).** EDSS, expanded disability status scale; MS, multiple sclerosis; SPMS, secondary progressive multiple sclerosis.

| Characteristic | SPMS Patients | SOs | Neurologists | Other HPs |
|---|---|---|---|---|
| | N = 15 | N = 7 | N = 7 | N = 12 |
| | N (%) | | | |
| Women | 8 (53.3) | 5 (71.4) | 6 (85.7) | 10 (83.3) |
| Age (years)[1] | 48.7 (39–60) | 43.0 (27–51) | 47.1 (32–49) | 48.6 (40–58) |
| Area of Italy | | | | |
| • North | 7 (46.0) | 6 (85.7) | 3 (42.8) | 7 (58.3) |
| • Center | 4 (27.0) | 1 (14.3) | 2 (28.6) | 3 (25.0) |
| • South | 4 (27.0) | | 2 (28.6) | 2 (16.7) |
| Education | | | | |
| • PhD/specialization | 0 | 2 (28.6) | | |
| • Degree | 7 (47.7) | 4 (57.1) | | |
| • High school | 5 (33.3) | 1 (14.3) | | |
| • Secondary school | 3 (20.0) | 0 | | |
| Occupation | | | | |
| • Employed | 12 (80.0) | 4 (57.1) | | |
| • Housewives | 0 | 2 (28.6) | | |
| • Unemployed | 0 | 1 (14.3) | | |
| • Retired (disability) | 2 (13.3) | 0 | | |
| • Retired (age) | 1 (6.7) | 0 | | |
| EDSS score[1] | 6.0 (4.0–7.0) | | | |
| Age at diagnosis of MS[1] | 35.7 (13–58) | | | |
| Time from SPMS diagnosis (years) | 2.2 (0.5–4) | | | |
| Working activity reduced | | | | |
| • No | | 4 (57.1) | | |
| • Partially | | 2 (28.6) | | |
| • Totally | | 1 (14.3) | | |
| Assistance provided | | | | |
| • Part of the day | | 5 (71.4) | | |
| • Part of the day and night | | 1 (14.3) | | |
| • All day long | | 1 (14.3) | | |
| Expertise in MS (years) | | | 17.4 (3–32) | 16.2 (1–39) |
| MS patients followed in the last 3 months | | | 165.7 (60–500) | 117.9 (5–250) |
| SPMS patients followed in the last 3 months | | | 25.0 (5–50) | 20.3 (5–70) |
| Profession | | | | |
| • Nurse | | | | 6 (50.0) |
| • Psychologist | | | | 3 (25.0) |
| • Physiotherapist | | | | 2 (16.7) |
| • Social worker | | | | 1 (8.3) |

1. Median (range)

To note, while SOs perceived the transition to SP as a sudden event *"[the transition to SPMS] it was quite sudden, not . . . not gradual. [SO03: wife, 42 years, North]"*, the other stakeholders described it as hard to be dated and defined because of its low evolution: *"It is very difficult to sanction this passage!" [HP05: woman, 50 years, social worker, North]*.

**Table 2. Theme 1: The awareness of the transition.**

| Categories | When | How | Due to... | Patient's interview style |
|---|---|---|---|---|
| Sub-categories | Patient's awareness:<br>• Before the communication (Ps, SOs, Ns)<br>• After the communication (P)<br>• Never happened, the patient is not aware (Ps, HPs) | • Suddenly (SOs)<br>• Difficulties in defining when the transition happened (Ps, Ns, HPs) | • Disease symptoms (P, SOs)<br>• Impact on activities of daily living/Loss of autonomy (Ps, SOs)<br>• No more remission (Ps, SOs)<br>• Worsening despite magnetic resonance imaging stability (Ps, SOs, Ns, HPs)<br>• Participation in this project (Ps, SOs)<br>• Changing or discontinuing disease modifying treatment (Ps, Ns, HPs) | • Tortuous (Is)<br>• Induce participant's answer (Is)<br>• Vague (Is)<br>• Verbose (Is) |

**Communication of the transition.** Data also showed a great variability in the dynamics around the communication of the transition (Table 3). It may be a progressive acquisition within the context of the patient-doctor relationship:

"*We concluded that the disease was turning from relapsing-remitting to SPMS. The doctor explained me SPMS is not connected to active plaques. More than anything else it was an observation . . . of both . . . because, as I told you, the situation slowed down . . . it got worse . . . and the resonance told you that the disease was stable, but it wasn't really the case from a clinical point of view. The EDSS passed form 5.5, to 6, and now we are at 7 . . .*" [P15: *man, 39 years, EDSS 7.0, South*].

However, for the majority the communication was ambiguous or absent and the topic was indirectly reported:

"*It is something implicit. It is hard that it is clearly faced. Sometimes, we [HPs] have some difficulties in stating that the patients have transitioned to a progressive form.*" [N04: *woman, 50 years, Center*]; "*They [HPs] did not tell you about what was going on. Nobody clearly states that you are entering in the progressive phase.*" [SO04: *wife, 49 years, North*].

Despite the diffused difficulties around this topic, all the stakeholders agreed that good communication of the transition is crucial.

"*Knowing it and being aware of the period I was going to live, receiving this information from an authoritative external source, would have given me a little more comfort, for sure!*" [P01: *man, 46 years, EDSS 6.5, North*]

"*I believe it is useful and people deserve to know about it.*" [N03: *woman, 49 years, Center*]

Data also revealed neurologists have different attitudes towards the communication of the transition to a progressive form. Someone believe it should be described as a possible outcome of the disease since the beginning: "*I usually explain at the moment of the diagnosis, that MS has an inflammatory phase and a degenerative phase*". [N07: *woman, 34 years, North*]. Other believe it should be introduced only later on:

"*I have some resistances to explain that there is 'phase A and phase B', immediately at the beginning. I do not agree so much with these classifications. It is obvious that there is a degenerative component, but when there is a 20-year-old girl in front of you, I believe this is not the*

**Table 3. Theme 2: Communication of the transition.** HP, health professional; MS, multiple sclerosis; N, neurologist; SO, patient significant other; SPMS, secondary progressive multiple sclerosis; RRMS, relapsing remitting multiple sclerosis.

| Categories | Dynamics | How communication is perceived | When neurologist speaks about SPMS | Patient's reaction to the communication |
|---|---|---|---|---|
| Sub-categories | • Acknowledgment with the neurologist (Ps, SOs, Ns, HPs)<br>• Acknowledgment after patients request for more information (P, SO, Ns)<br>• Indirect (i.e. medical report) (Ps, HP)<br>• Prevented by the patients (Ps)<br>• Progression is not communicated (Ps, SOs)<br>• Ambiguous/unclear (P, SO, N, HP)<br>• Open and clear (SO, HP) | • Important (P, SOs, Ns, HPs)<br>• Not important (Ps, Ns) | • Since the diagnosis of RRMS (N)<br>• Not at the moment of the diagnosis, but further on (N) | • Emotional<br> - Panic (P)<br> - Fear (P)<br>• Defensive strategies<br> - denial (P, SO, N)<br> - avoidance (Ps,N)<br> - displacement (P)<br>• Openness to experience (P)<br>• Acceptance (P) |

*main topic! You work on that further on and with the adequate time".* [N01: man, 56 years, North]

On the other hand, also patients found it difficult to speak about the transition as demonstrated by the massive use of defensive strategies (i.e. denial, avoidance, displacement).

*"I knew it [transition to SPMS]. I sensed something. . . and the doctor told me about it, but the reality is that I didn't want to understand it, to face the reality. . . I can't even speak about it!"* [P05: woman, 44 years, EDSS 6.0, North]

*"When the neurologist invited my husband to participate in this study, the first reaction was: It is a mistake! I will write him that he is wrong, it is not true!"* [SO01: wife, 48 years, North]

**Dealing with symptoms worsening.** People transitioning to SPMS showed the tendency to speak about their progression in terms of symptoms worsening more than a change in functioning (Table 4). Being in contact with the experience of worsening activates a lot of emotions and feelings (such as anxiety, anger, guilt, sadness, loneliness) and consumes lots of energy:

*"This is hard! To keep staying in balance! Sometimes you feel like a swimmer that treads water in the middle of a lake! You can't keep going anymore, but you still keep saying: I can't drown. So you keep treading water illogically, without any purpose, putting all your energy in this scrappy way! You keep doing that because it let you float. If you stop, if you let yourself have a break, it means you die, you drown!"* [P06: woman, 39 years, EDSS 7.0, North] (Table 4).

In facing worsening, patients showed a variety of adjustment strategies that can be collected in four groups: concrete (problem-focused) solutions, inner resources, spirituality and social connectedness.

Some participants reported that concrete solutions as looking for information, maximizing autonomy in everyday functioning, work maintainance, self-care activities, and staying in contact with the HPs, helped.

*"I usually surf the web, read the forum, I do my best to keep me informed about new drugs. . ."* [P11: man, 47 years, EDSS 6.5, Center];

*"I have always tried to keep my job, to never leave it, because it is a very important aspect of my life. My job is my anchor to the reality, to the person I was before."* [P01: man, 46 years, EDSS 6.5, North]

**Table 4. Theme 3: Dealing with symptoms worsening.** HP, health professional; N, neurologist; SO, patient significant other.

| Categories | Personal Experience | Adjustment strategies | Difficulties in adjustment |
|---|---|---|---|
| Sub-categories | • Anxiety *(P)*<br>• Anger *(P, SOs)*<br>• Guilt *(P, SO)*<br>• Fear *(P, SO, HP)*<br>• Sadness *(Ps)*<br>• Loneliness of both patients and SOs *(Ps, SOs)*<br>• Humiliation/shame *(Ps, HPs)*<br>• Disappointment *(P, SO)*<br>• Helplessness/powerlessness *(SO, Ns, HPs)*<br>• Nervousness of both patients and SOs *(SOs)*<br>• Frustration *(Ns)*<br>• Hope *(HP)*<br>• Confused/unprepared *(SO; HPs)*<br>• Worry *(HP)*<br>• Shame *(HP)* | *Concrete solutions*<br>• Surfing the web *(Ps, SO)*<br>• Looking for new drugs *(Ps, SO)*<br>• Autonomy maintenance *(Ps, SO)*<br>• Work maintenance *(P, SOs)*<br>• Planning activities carefully (one activity at time) *(Ps)*<br>• Self-care *(Ps)*<br>• Staying in contact with the neurologist or HPs *(P, SO, N, HP)*<br>*Inner resources*<br>• Tenaciousness *(P)*<br>• Grounding *(P)*<br>• Acceptance *(Ps)*<br>• Meaning *(P, SO)*<br>• Mindfulness (carpe diem) *(SO)*<br>• Being in contact with personal values *(HPs)*<br>*Spirituality*<br>• Religion/faith *(P)*<br>*Social connectedness*<br>• Social support *(Ps)* | *Inner factors*<br>• Giving up *(Ps, HP)*<br>• Not sharing *(Ps)*<br>• Retirement from the relationship *(P)*<br>• Cognitive fusion *(P)*<br>• Passiveness *(HP)*<br>• Experiential avoidance/denial *(P, SO)*<br>*Environmental factors*<br>• Uncertainty *(P, SO)*<br>• Presence of children *(SOs)* |

Other emphasized the importance of cultivating certain inner resources, such as grounding, present moment awareness (mindfulness), acceptance, finding a meaning and being in contact with personal values.

> *"What can I do if I need to go from A to B and I am encountering some barriers? I can be stuck somewhere, or I can accept it and seek a way to overcome it, maybe walking around the 'stone', going back and taking another road, finding other strategies. I know I have the tendency to avoid but, in the end, you have to deal with what is going on." [P07: man, 49 years, EDSS 6.0, North]*

Spirituality was also another important resource to deal with MS worsening: *"I have found some relief in religion; I always had a good relationship with God. Church has been my only outlet." [P01: man, 46 years, EDSS 6.5, North];* as well as social support: *Having many friends helped me a lot! Telling them about my situations and emotions helped me a lot! [P06: woman, 39 years, EDSS 7.0, North].*

Besides the several resources activated, narratives were also rich in clear signs of maladjustment, such as giving up, not sharing, retirement from relationships, cognitive fusion.

> *"I withdrew from the world. My life is now the PC, the TV and playing with crosswords. Sometimes my daughter visits me but it is no more the same. Also about meeting other people, it is so rare!" [P03: man, 52 years, EDSS 6.0, North]*

> *"Being passive, we should work with patients on this aspect." [HP06: man, 46 years, psychologist, North]*

Uncertainty and presence of children where also listed as aspects that can further complicate the situation.

> *"This disease is really unpredictable! This unpredictability and uncertainty drives me crazy! I would prefer to know he will be on a wheelchair and know his future limitations than staiyng in this uncertainty." [SO01: wife, 48 years, North]*

**Table 5. Theme 4: Met and unmet needs.** AISM, Italian Multiple Sclerosis Society; HP, health professional; MS, multiple sclerosis; N, neurologist; SO, patient significant other; SPMS, secondary progressive multiple sclerosis; RRMS, relapsing remitting multiple sclerosis.

| Categories | Met needs | Unmet needs |
|---|---|---|
| Sub-categories | *Organization and management*<br>• Multidisciplinary equipe/holistic approach *(N, HP)*<br>• MS specialists (i.e. urologist, psychologist) *(Ns, HPs)*<br>• Neurologist availability *(Ps, SO, N)*<br>• Physiotherapy *(P, SO, N, HP)*<br>• Connection with AISM *(P, SO, N, HP)*<br>*Empowerment Training*<br>• Patients training (i.e. nutritionist, how to deal with daily life, hints for searching information) *(N)*<br>*Information*<br>• Easy access to information about aids *(P, SO)* | *Organization and management*<br>• Ad hoc plan of follow up visits with the neurologists and other HPs *(P, Ns, HP)*<br>• Multidisciplinary care/holistic approach *(P, SO, Ns, HPs)*<br>• Patient and SOs participation in multidisciplinary care *(SOs, Ns, HPs)*<br>• A complete service charter of the MS center *(HP)*<br>• Home visits/care *(P)*<br>• Fast and easy access to MS center and its professionals *(P, SO, N, HP)*<br>• A case manager *(Ps, SOs, HPs)*<br>• Rehabilitation: More physiotherapy/occupational therapy *(Ps, SOs, Ns)*<br>• Collaboration with the general practitioner *(HPs)*<br>• Connection with AISM *(SOs)*<br>• Psychological support for both patients and SOs (individual, couple, family or group setting; homogeneous (SPMS) group discussion for both patients and relatives *(Ps, SOs, Ns, HPs)*<br>• Self-help group (social networking; communication beyond self-help group e.g. MS café; "Happy hour for sharing" experience (only SOs permitted) *(P, SOs)*<br>• Psychological support for HPs *(Ns)*<br>*Empowerment Training*<br>• Patients (i.e nutritionist, how to deal with daily life, hints for searching information) *(Ns)*<br>• HPs<br> - Communication *(Ps, N)*<br> - National and international cross training on SPMS (Meetings with multidisciplinary team to learn how to deal with daily living activities—evidence based, up to date information about symptomatic treatments; lifestyle, disease modifying treatments) *(SOs, Ns, HPs)*<br>• SOs (i.e. disease and patient management; how to deal with daily life; how to deal with children in case one of the parents has SPMS) *(P, SOs, Ns)*<br>*Information*<br>• Information on SPMS for patients and SOs *(Ps, SOs)*<br>• Clear information on non-pharmacological approaches and treatment/lifestyle/aids *(P, SOs, N, HPs)*<br>• Social care information (i.e. L.104) *(SO, N)*<br>• Website (or other sources) to share solutions and information *(P, SOs)*<br>*Policies*<br>• Job outplacement *(P, SOs)*<br>• Improving the social security net *(Ps, HPs)*<br>• Facilitating bureaucracy *(P, SO)*<br>• Supporting patient's autonomy/accessibility *(P, SO, HPs)*<br>• Developing specific national guidelines for people that are experiencing transition from RR to SPMS *(HP)* |

*"You really feel powerless; you don't know what to do! It is not only about my husband, I also have to think about my son! When I think about all this, I really have the feeling my word is falling apart!" [SO03: wife, 42 years, North]*

**Needs.** Data reported in Table 5 provide an overview of met and unmet needs of people transitioning to SPMS, as reported by the different stakeholders. Needs were grouped into 'organization and management'; 'empowerment training'; 'information' and 'policies' (this latter was detected only within the unmet needs category).

**Met needs.** There is no homogenous answer to patients' needs among the participants, but it is relevant that some of them reported the following needs as already met: a good connection with the neurologist and patient's association, an easy access to physiotherapy, and other MS specialists' consultation/treatment.

*AISM [Italian MS Society] has been a great discovery! It is not only about fundraising, they really help you! [P04: man, 44 years, EDSS 6.0, North]*

*We work with many people with SPMS, physiotherapy is very often part of a multidisciplinary approach. Enhancing and promoting participation is one of our goals, also with people with SPMS. [HP11: man, 45 years, physiotherapist, North]*

The availability of training courses for patients was also reported as a precious opportunity:

*We organize monthly meeting on a variety of topics such as nutrition and lifestyle! [N02: woman, 59 years, North]*

An easy access to information about aids available was crucial for two participants. Unfortunately, it was the result of their proactiveness and curiosity and not of a structured resource available for other patients:

*"Because of my job, I travel a lot and I can easily have access to more information about which aids are used in different country. We found this amazing aid that helps her walking and that it is not used in Italy! On the contrary, it was very easy to find it in the UK." [SO02: husband, 41 years, North].*

**Unmet needs.** Participants reported several and shared unmet needs. They called for many organizational changes and all the four stakeholders stood for the importance of implementing a holistic and multidisciplinary approach, increasing the availability/responsiveness of the MS center and offering psychological support to both patients and SOs.

*"What we need is a holistic approach." [SO04: wife, 49 years, North]*

*"We need teambuilding. The team must be really connected, not only when you are hospitalized, not only inside the Center, but in general. What matters is that there is a real dialogue between the various players. That is absent, particularly between providers from different centers (silos working, lack of flexibility). We also need to dialogue with the patient and take into account the patient's culture and capabilities." [HP05: woman, 50 years, social worker, North]*

Patients and SOs should be part of the multidisciplinary team and contribute to the development of an ad hoc plan of follow up visits with the neurologists and other HPs.

*"These patients should have a dedicated path, which differs from the outpatient performance. If I prescribe my patient a consultation with a physiatrist, it can take up to six or seven months with the public healthcare system." [N07: woman, 34 years, North]*

An easiest and continuous access to rehabilitation was also reported as an important unmet need.

*"The healthcare system provides them with only two rehab courses per year! However, these patients would benefit from a more continuous program. They ask us how they could do, but we have no answer!" [N07: woman, 34 years, North]*

All the stakeholders envisaged the need for improving their knowledge and skills. Training may cover different topics, such as lifestyle for patients, communication for HPs, and disease and patient management for SOs.

*"I think that we should also organize meetings for patients on daily life management." [N04: woman, 50 years, Center]*

*"We must improve our knowledge, know how to do it [communicate the transition] better. And we need tools that help us to communicate, definitely a crucial step. As for the communication of the MS diagnosis, where, thanks to the support of different tools, we can give the patient a perspective: here [communication of SPMS transition] symptomatic therapies, rehabilitation, and . . . the capability to organize a new strategy of action." [N03: woman, 49 years, Center]*

*"Training, training. That is, we need to train HPs so that they are able to act at their best. But if they are blockheads . . . there is nothing to do." [HP02: woman, 48 years, nurse, North]*

*"There is absolutely no instruction manual. But on one thing I agree with what you said, that we need instructions for us. There is no instruction manual, absolutely. I give you an example: they prescribed the wheelchair, choosing it without considering my wife needs, and now we are having the changes to the chair on our own pocket. Another example: getting on the car. A trivial procedure for every person, which is done every day, twenty times a day. We do not notice it. With my wife it is the hell. Because you do not know exactly how to take her, how to put her in, how to turn her, then she gets angry. . . in short, it is the slaughter that we all know." [SO02: husband, 41 years, North]*

In line with the needs for training, information is another important area of interest that was quoted by all the four stakeholders. It covers the following topics: SPMS, non-pharmacological approaches, and treatment/lifestyle/aids, and social care (i.e. L.104).

*"A kind of booklet. . . the disease has changed, maybe I need psychological support, where can I go? Try to provide tracks that one can walk." [P06: woman, 39 years, EDSS 7.0, North]*

*"What I'm missing, as also pointed out by the other participant, is a . . . another type of catalogue, let's say. A catalogue with a compendium of all possible and imaginable aids. We recently discovered a . . . a wheelchair with BMX wheels, double, to go to the beach." [SO07: husband, 42 years, South]*

*"I discovered the law 104 only two years ago, while we could have benefited from it since long. Nobody informed us about this before, I knew it from the labor unions." [SO01: wife, 48 years, North]*

At a wider level, stakeholders asked for new policies able to facilitate job outplacement, improve social security net, reduce bureaucracy, and support people autonomy.

*"In the end, eh . . . he got along with his boss, who said: 'Don't worry. Stay at home, take care of yourself, take your treatments. . .' but they fired him! And after that he got worse, just saying: 'Now how . . .? What can I do?'" [SO05: wife, 51 years, North]*

*"In my opinion, in Italy, we are really in a bad situation as we lack the sense of 'welfare'. Last summer I went to England, and going around the city was not a problem: there is the disabled service you call and, if you book early, take you where you want in the city. You need, let's say*

*the shopping at home? You call them, they bring the shopping to you. Hmm . . . I wanted to go to London alone by train. I said, "Ok, now you've done 30, you do 31." This is what we really miss: services at a social level that make everything easy." [P06: woman, 39 years, EDSS 7.0, North]*

## Discussion

This qualitative study describes the experience of the transition to SPMS, as reported by different stakeholders—people with SPMS, their SOs, neurologists and HPs, living and working in Italy. This is the first study exploring this topic in a country different from the UK, and using mixed methods [6–10].

Two are the main findings: first, the widespread lack of awareness around the transition; second, the need to improve the quality of the care pathway for patients living the transition to SPMS and their families.

Lack of awareness about the transition to SPMS was surprisingly recurring among the patients who participated in the interviews. This result was also confirmed by the online survey that followed, reporting an unawareness rate of 43% within the Italian participants [21]. The qualitative study suggests some explanations for this lack of awareness. On one side, patients may have the tendency to deny the transition, because of the high emotional burden. On the other side, clear difficulties in the patient-doctor communication appeared. In fact, in few cases the communication of the transition was a shared process, more often it was described as indirect (written down on medical report, but never directly told to the patient) or unclear. An upstream factor that contributes to communication problems is the intrinsic difficulty in defining when the transition happened. This theme was reported by neurologists and HPs during the FGMs, and it is also well known in the literature, where an uncertainty period of about two years has been reported between neurologist's first suspicion and confirmation of SPMS conversion [5, 22]. Another important factor is the neurologist's attitude towards the topic of the transition and towards SPMS. In fact, some HPs deemed the communication of the transition as important, while others believed it may not be useful; some introduced the topic of the progressive form since the beginning of patient care, others preferred to wait. In particular, neurologists may be reluctant to recognize the bad prognosis and the scarcity of effective disease-modifying treatments for this disease form. As a result, they may avoid having conversations on these topics with patients and their families. Our qualitative data are in line with the results of the study run by O'Loughlin and colleagues, in which the process of moving from uncertainty toward confirmation of patient's diagnostic label was moderated by HPs attitudes and approaches [9].

The complexity of this dynamic can also explain the mismatch observed by Solari et al. in the online survey. While patient-neurologist agreement on age at MS diagnosis was very high, patients reported age at SPMS conversion 2.7 years lower on average than neurologists [21], suggesting that there is a certain amount of time in which the patient is alone in dealing with the signs of the transition and her/his own worries and thoughts. All these aspects may amplify the risk of feelings of helplessness, a lack of directions for symptom management, and lack of support from the healthcare system already detected in a recent longitudinal study on the transition to SPMS [10]. The request for a training to improve communication skills spontaneously arisen by neurologist during this study can be a first step towards a better management of this delicate communication process. As reported in the qualitative study, the transition to SPMS may change the interaction dynamics between persons with SPMS, SOs and HPs, particularly the neurologist. In fact, the suspension of disease-modifying drugs may leave an "empty space"

in the caring relationship. The sensation activated by this "uncovered place" may be a new drive for neurologists for topping up their psychological skills, so precious in effectively supporting people with chronic diseases. It may also suggest the need for a closer collaboration between neurologists and psychologists in the demanding task of supporting people with SPMS and their family.

The experiences and views of the four parties converged and complemented each other, the only contrast being about the perception of the transition as a sudden (SOs) or slow process (patients, neurologists and other HPs).

Our data helped outline the future directions to improve people with SPMS and their SOs' experience of the transition. It involves a multi-levels approach concerning organization and management of the patient's care pathway, empowerment trainings, information availability, and policies. In 2015, Davies et al. [6] described the frustration of patients and carers towards the health and social care system in the UK. The stakeholders involved in our study also reported the need for changes at both organizational and policy levels. Participants depicted an ideal model for clinical practice with people transitioning to SPMS. Patient (and SOs) should participate in the multidisciplinary care planning, to ensure it is a personalized care plan, as supported by Coulter et al. [22]. The process should be supervised by a case manager (patient's privileged speaker) able to act as a bridge between the patient (and her/his SOs), the HPs and the services eventually needed and available in the territory. In fact, individualized relationship between patients and HPs and a tailored communication are associated with higher patients' satisfaction with healthcare [23].

Coherently with Davies et al. findings [7], psychological support and intervention aimed at promoting social connectedness for both patients and SOs should also be implemented. The need for psychological support for patients and their SOs is well established by both the maladjustments signs detected in the interviews (i.e. giving up, not sharing, retirement from the relationship, cognitive fusion, passiveness, and experiential avoidance/denial) and stakeholders unanimous request for specialized psychological interventions. We expect that the availability of psychological support in the patient care pathway may improve patients and SOs satisfaction with the center, as supported by a recent study [23].

Another important potential future direction is the development of tailored training for each of the four stakeholders. This is coherent with the transition theory principle: to deal better with transition, the person requires knowledge about what to expect and what strategies he/she might find helpful [24].

Finally, stakeholders reported the need for changes at a policy level. In particular, they advocate for initiatives to support job maintenance in people with SPMS. As reported in the literature, job loss brings detrimental consequences to patients and their families concerning short-term and long-term economic, psychosocial and healthcare utilization domains [25]. Moreover, job loss has been associated with worse self-reported health and quality of life and increased adverse health behaviors [26]. Other policy issues concern increasing accessibility, in order to support or prolong patient's autonomy and reduced bureaucracy. The latter is strictly connected with the unmet need of receiving more social care information.

## Limitations

Despite the efforts to comprehensively capture the experience of patients transitioning to SPMS, their SOs, neurologists and HPs, it is possible that some aspects may have been missed. Particularly because SOs were mainly from the North of Italy and none of them had a relative with SPMS with severe cognitive compromise. We cannot exclude that more variability in the SOs sample would have enriched the results. As with all qualitative analyses, these results can

be applied to the context (e.g. Italy) in which the study was conducted. The transfer of the results to other countries and environments must be handled with caution.

## Conclusions

To our knowledge, this is the first study that explored the experiences of the transition to SPMS outside the UK and taking into consideration the perspective of the different stakeholders. Within the context of the qualitative approach, study results provide with an overview of the difficulties in dealing with the transition to SPMS and outlined which changes should be implemented.

The diffuse lack of awareness around the transition is a first important warning sign of the difficulties experienced by all the stakeholders during this delicate phase of the disease. Tailored interventions (psychological intervention, educational and information trainings) to support people with SPMS and their family while facing the transition, and to train HPs in communication skills are essential for the future improvement of the quality of care. In terms of reorganization of the care pathway in the Italian context, it was particularly stressed the need for a holistic and multidisciplinary approach (with patients and SOs as members of the team), and the development of an ad hoc plan of follow up visits with easy access to MS specialists' counsultation/treatment. Changes at policy level were also requested particularly in order to safeguards people with MS working activities, improve accessibility and facilitate bureaucracy.

A dedicated paper reporting the results of the qualitative study performed in Germany within the ManTra project is under preparation. Further studies should explore this topic in other countries taking into consideration the specificities of each context in terms of policies, health care system, and culture. It could also be an opportunity to evaluate weakness and strengths of different realities and promoting good practices in Europe.

## Supporting information

**S1 Appendix. COREQ.**
(PDF)

**S2 Appendix. PSY and FMG guides.**
(PDF)

**S3 Appendix. Audit trail.**
(PDF)

## Acknowledgments

The authors are grateful to all persons living with MS, their significant others, and health care professionals who participated in this study.

**ManTra project investigators:** Steering Committee: CB; P Confalonieri - Fondazione IRCCS Istituto Neurologico C Besta, Milan, Italy; G De Luca - G d'Annunzio University of Chieti-Pescara, Chieti, Italy; AG,; AMG (study Co-PI); Laura Gitto– Università degli Studi di Roma 'Tor Vergata', Rome, Italy; CH - University Medical Center Hamburg-Eppendorf, Hamburg, Germany; AS (study PI); V Torri Clerici - Fondazione IRCCS Istituto Neurologico C Besta, Milan, Italy; M Trojano - University of Bari, Bari, Italy; M Messmer Ucelli - Italian Multiple Sclerosis Society and Research Foundation (AISM), Genoa, Italy.

Literature review panel: A Fittipaldo- Fondazione IRCCS Istituto Neurologico C Besta, Milan, Italy; S Köpke—University of Lubeck, Lubeck, Germany; AMG; AG.

Qualitative analysis panel: CB, AB, AMG, EP.

Psychologist working group: Five psychologists with expertise in MS who participate in a training on qualitative research focused on the ManTra project—M Falautano and E Minacapelli, San Raffaele Scientific Institute, Milan, Italy; R Quintas and C Scaratti, Fondazione IRCCS Istituto Neurologico C Besta, Milan, Italy; R Sartori, University of Milan, Milan, Italy.

Expert panel: AG, AMG, LG, EP, MMU, VTC, C Tortorella—San Camillo Forlanini Hospital, Rome, Italy.

Centers and investigators: Fondazione IRCCS Istituto Neurologico C Besta, Milan: Unit of Neuroepidemiology, AS, AG, G Ferrari, AF; Unit of Neuroimmunology and Neuromuscular Diseases: PC, AMG, VTC, R Mantegazza; Department of Neuroscience, Imaging and Clinical Sciences, G d'Annunzio University of Chieti-Pescara, Chieti: M Onofrj, EP, D Farina, D Travaglini, G De Luca; Department of Neurosciences, San Camillo Forlanini Hospital, Rome, Italy: CT, ME Quartuccio; Departments of Basic Medical Sciences, Neurosciences and Sense Organs, Aldo Moro University of Bari, Bari: MT, E Luciannatelli, R Viterbo; Institute of Neuroimmunology and Multiple Sclerosis, University Medical Center Hamburg-Eppendorf, Hamburg, Germany: IS, AB, CH

## Author Contributions

**Conceptualization:** Ambra Mara Giovannetti, Andrea Giordano, Alessandra Solari.

**Data curation:** Ambra Mara Giovannetti, Alessandra Solari.

**Formal analysis:** Ambra Mara Giovannetti, Erika Pietrolongo, Claudia Borreani.

**Funding acquisition:** Ambra Mara Giovannetti, Andrea Giordano, Alessandra Solari.

**Investigation:** Ambra Mara Giovannetti, Alessandra Solari.

**Methodology:** Ambra Mara Giovannetti, Andrea Giordano, Alessandra Solari.

**Project administration:** Alessandra Solari.

**Resources:** Alessandra Solari.

**Supervision:** Christoph Heesen, Alessandra Solari.

**Validation:** Alessandra Solari.

**Visualization:** Ambra Mara Giovannetti, Alessandra Solari.

**Writing – original draft:** Ambra Mara Giovannetti, Erika Pietrolongo, Andrea Giordano, Insa Schiffmann, Anna Barabasch, Christoph Heesen, Alessandra Solari.

**Writing – review & editing:** Ambra Mara Giovannetti, Erika Pietrolongo, Andrea Giordano, Insa Schiffmann, Anna Barabasch, Christoph Heesen, Alessandra Solari.

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
