## [Decision Letter · Decision Letter 0]

31 Dec 2019

PONE-D-19-27992

Conversion to secondary progressive MS: Multistakeholder experiences and needs in Italy

PLOS ONE

Dear Dr. Solari,

Thank you for submitting your manuscript to PLOS ONE. After careful consideration, we feel that it has merit but does not fully meet PLOS ONE’s publication criteria as it currently stands. Therefore, we invite you to submit a revised version of the manuscript that addresses the points raised during the review process.

We would appreciate receiving your revised manuscript by Feb 14 2020 11:59PM. To enhance the reproducibility of your results, we recommend that if applicable you deposit your laboratory protocols in protocols.io, where a protocol can be assigned its own identifier (DOI) such that it can be cited independently in the future. For instructions see: http://journals.plos.org/plosone/s/submission-guidelines#loc-laboratory-protocols

We look forward to receiving your revised manuscript.

Kind regards,

Luigi Lavorgna

Academic Editor

PLOS ONE

Journal Requirements:

2. Please include captions for your Supporting Information files at the end of your manuscript, and update any in-text citations to match accordingly. Please see our Supporting Information guidelines for more information: http://journals.plos.org/plosone/s/supporting-information

Reviewers' comments:

Reviewer's Responses to Questions

**Comments to the Author**

1. Is the manuscript technically sound, and do the data support the conclusions?

Reviewer #1: Yes

Reviewer #2: Partly

2. Has the statistical analysis been performed appropriately and rigorously? 

Reviewer #1: Yes

Reviewer #2: N/A

3. Have the authors made all data underlying the findings in their manuscript fully available?

Reviewer #1: Yes

Reviewer #2: Yes

4. Is the manuscript presented in an intelligible fashion and written in standard English?

Reviewer #1: Yes

Reviewer #2: Yes

5. Review Comments to the Author

Reviewer #1: Solari and colleagues reported on the impact of transition to secondary progressive multiple sclerosis. This is very interesting topic, which covers a difficult moment of disease. Insights from this study can be helpful to healthcare professional for better patients’ management. The manuscript is overall clear and well written. Authors provided full data access and complied with research guidelines. However, I have raised some issues I would like the authors to address.

I do not understand why in the introduction authors mention also the German chapter of this research, but, then, did not report on this. Are there any methodological differences in study conduction between Germany and Italy justifying two different papers based on the same project? Otherwise, I do expect to get presented both. It would be very much helpful also comparing Germany and Italy to have better insight into this issue.

The paragraph “In line with the above studies… before SPMS was medically confirmed” should be moved from the Introduction to methods and results, as appropriate. Also, I would move the paragraph “Managing the transition to SPMS… newly diagnosed patients with SPMS” at the very end of the introduction, since it contains study objectives.

A key point of this manuscript is the lack of awareness of patients and the variability of neurologists’ attitude towards SP conversion. As a matter of fact, a diagnosis of SPMS limits the possibility of DMT prescription and, more in general, of room for improvement in MS management. These could be responsible at least in part for the delay in officially identifying SPMS in neurology clinical practice.

In the limitations’ section of the discussion, when discussing on the SOs sample, I would suggest authors refer to Neate et al Plos One 2019. Also, SP patients suffer from cognitive impairment at some level (Moccia et al. Mult Scler 2016), and formal testing would have been helpful.

Reviewer #2: The authors investigate a relevant topic in the field of progressive MS, particularly concerning transition to secondary progressive MS and its significance for patients, their significant others, neurologists and other health professionals.

I have no major concerns about the manuscript, as the authors correctly explain the complexity of this research field particularly when dealing with qualitative measures of psychological variables and they clearly report research findings.

Minor points:

- since lack of SPMS awareness in patients resulted as one of the major findings it would be interesting to understand whether aware and unaware MS patients showed any relevant difference in terms of the two themes "Dealing with symptoms worsening" and "Needs";

- authors should explain more in detail study limitations regarding low sample size - particularly in subgroups - and lack of a standardized definition of SPMS among other factors.

6. PLOS authors have the option to publish the peer review history of their article (what does this mean?). If published, this will include your full peer review and any attached files.

Reviewer #1: No

Reviewer #2: Yes: Alberto Gajofatto

---

## [Author Response · Author response to Decision Letter 0]

16 Jan 2020

Responses to the reviewers

Reviewer #1

1. I do not understand why in the introduction authors mention also the German chapter of this research, but, then, did not report on this. Are there any methodological differences in study conduction between Germany and Italy justifying two different papers based on the same project? Otherwise, I do expect to get presented both. It would be very much helpful also comparing Germany and Italy to have better insight into this issue. We thank the reviewer for his comment. As from the published protocol (manuscript’s reference 12), the ManTra project originated in Italy and it was then adopted in Germany. Concerning the qualitative study (ManTra project action 2), the eligibility criteria and the interview guides were the same in the two countries. However, for feasibility reasons, the German protocol differed in the following: a) patient personal semistructured interviews (PSIs) were replaced by telephone PSIs in Northern Germany, and by a focus group meeting (FGM) in Southern Germany; b) patient significant other PSIs were replaced by a FGM; c) Instead of two FGMs (one with neurologists, and one with other health professionals) there was one FGM with neurologists and other health professionals. We agreed to present the findings of this project action separately in view of these differences, and also because the analysis was run separately in the two countries by Italian and German researchers in order not to miss any conceptual and semantic nuance. Reporting all the process in a single manuscript would have been too much, even in a journal that has not space constraints. In the following manuscript, we will describe the German qualitative study, consider contrasts and similarities of findings across the two countries, and highlight the need of a guidance document for qualitative analyses of data from different languages/cultures. 

2. The paragraph “In line with the above studies… before SPMS was medically confirmed” should be moved from the Introduction to methods and results, as appropriate. Also, I would move the paragraph “Managing the transition to SPMS… newly diagnosed patients with SPMS” at the very end of the introduction, since it contains study objectives. The manuscript reports the results of one action (action 2, Italian findings) of the ManTra project. For this reason, in the Introduction we included a description of the project at large. However, the present point and the following one indicate that we were not clear enough in our description. Thus, following the reviewer’s advice we have re-shaped the Introduction by outlining more briefly the ManTra project, at the end of the Introduction, specifying in the last sentence the objective of the present paper. 

3. A key point of this manuscript is the lack of awareness of patients and the variability of neurologists’ attitude towards SP conversion. As a matter of fact, a diagnosis of SPMS limits the possibility of DMT prescription and, more in general, of room for improvement in MS management. These could be responsible at least in part for the delay in officially identifying SPMS in neurology clinical practice. We agree with the reviewer that the limited availability of DMT for SPMS is one reason for postponing the disclosure of SP conversion (see e.g. Discussion: ‘In particular, neurologists may be reluctant to recognize the bad prognosis and the scarcity of effective disease-modifying treatments for this disease form. As a result, they may avoid having conversations on these topics with patients and their families.’).

4. In the limitations’ section of the discussion, when discussing on the SOs sample, I would suggest authors refer to Neate et al Plos One 2019. The paper by Neate et al. is an interesting one on the experiences of partners of people with MS who attended a healthy lifestyle workshop. Nevertheless, this study does not refer to our difficulty in enrolling significant others (SOs) from the South of Italy, and SOs of SPMS patients with severe cognitive compromise. 

5. Also, SP patients suffer from cognitive impairment at some level (Moccia et al. Mult Scler 2016), and formal testing would have been helpful. We agree that cognitive compromise is common in SPMS. To safeguard the validity of our findings, patients with severe cognitive compromise (referring neurologist judgment) or unable to communicate effectively were excluded. We purposely avoided to exclude patients with mild or moderate cognitive impairment, and deemed that a formal cognitive testing was not necessary. Notably, we did not include patients whose SPMS conversion dated over the preceding 5 years. 

Reviewer #2

1. Since lack of SPMS awareness in patients resulted as one of the major findings it would be interesting to understand whether aware and unaware MS patients showed any relevant difference in terms of the two themes "Dealing with symptoms worsening" and "Needs". Participants who were unaware of the transition were less informative and vaguer in their narratives. Therefore, we cannot be sure that differences in the categories reported by the two subgroups are due to different experiences or are simply the results of the difficulties of the unaware patients in speaking about the topic. Moreover, we cannot guarantee that the saturation criterion is respected if we run separate analysis for aware/unaware patients.

On pages 13-14 below we have re-organized the two themes "Dealing with symptoms worsening" and "Needs" based on patient awareness. Taking into consideration the limitations reported above these results are only exploratory, and we prefer not to include them in the manuscript. Common subcategories for unaware and aware patients are reported in red. 

2. Authors should explain more in detail study limitations regarding low sample size - particularly in subgroups - and lack of a standardized definition of SPMS among other factors. In the Discussion, we have acknowledged among the study limitations the restricted number of SOs from the South of Italy, and the lack of SOs of patients with severe cognitive compromise. Nevertheless, any subgroup analysis is out of the scope of qualitative analysis. Regarding the SPMS criteria, we used Lublin 1996 (reference 2 of the manuscript).

---

## [Editor Report · Decision Letter 1]

21 Jan 2020

Conversion to secondary progressive multiple sclerosis: Multistakeholder experiences and needs in Italy

PONE-D-19-27992R1

Dear Dr. Solari,

We are pleased to inform you that your manuscript has been judged scientifically suitable for publication and will be formally accepted for publication once it complies with all outstanding technical requirements.

With kind regards,

Luigi Lavorgna

Academic Editor

PLOS ONE
---

## [Editor Report · Acceptance letter]

3 Feb 2020

PONE-D-19-27992R1 

Conversion to secondary progressive multiple sclerosis: Multistakeholder experiences and needs in Italy 

Dear Dr. Solari:

I am pleased to inform you that your manuscript has been deemed suitable for publication in PLOS ONE. Congratulations! Your manuscript is now with our production department. 

With kind regards,

on behalf of

Dr. Luigi Lavorgna 

Academic Editor

PLOS ONE